# A Highly Sensitive XNA-Based RT-qPCR Assay for the Identification of ALK, RET, and ROS1 Fusions in Lung Cancer

**DOI:** 10.3390/diagnostics14050488

**Published:** 2024-02-24

**Authors:** Bongyong Lee, Andrew Chern, Andrew Y. Fu, Aiguo Zhang, Michael Y. Sha

**Affiliations:** DiaCarta Inc., 4385 Hopyard Rd., Suite 100, Pleasanton, CA 94588, USA

**Keywords:** lung cancer, fusion, ALK, RET, ROS1, XNA, RT-qPCR

## Abstract

Lung cancer is often triggered by genetic alterations that result in the expression of oncogenic tyrosine kinases. Specifically, ALK, RET, and ROS1 chimeric receptor tyrosine kinases are observed in approximately 5–7%, 1–2%, and 1–2% of NSCLC patients, respectively. The presence of these fusion genes determines the response to tyrosine kinase inhibitors. Thus, accurate detection of these gene fusions is essential in cancer research and precision oncology. To address this need, we have developed a multiplexed RT-qPCR assay using xeno nucleic acid (XNA) molecular clamping technology to detect lung cancer fusions. This assay can quantitatively detect thirteen ALK, seven ROS1, and seven RET gene fusions in FFPE samples. The sensitivity of the assay was established at a limit of detection of 50 copies of the synthetic template. Our assay has successfully identified all fusion transcripts using 50 ng of RNA from both reference FFPE samples and cell lines. After validation, a total of 77 lung cancer patient FFPE samples were tested, demonstrating the effectiveness of the XNA-based fusion gene assay with clinical samples. Importantly, this assay is adaptable to highly degraded RNA samples with low input amounts. Future steps involve expanding the testing to include a broader range of clinical samples as well as cell-free RNAs to further validate its applicability and reliability.

## 1. Introduction

Lung cancer is one of the most common types of cancer and stands as the leading cause of tumor-related fatalities, accounting for nearly a quarter (approximately 25%) of such cases [1]. According to estimates, there were 117,560 new lung cancer cases in men and 120,790 in women in 2023. The estimated deaths due to lung cancer were 67,160 in men and 59,910 in women in 2023 [2]. Over 80% of lung tumors are classified histologically as either adenocarcinomas, squamous cell carcinomas, or large cell carcinomas, collectively categorized as non-small-cell lung cancers (NSCLCs). Various genetic alterations have been identified as drivers of oncogenic processes in NSCLC. These alterations encompass point mutations, deletions, insertions, and gene fusions. In Western nations, approximately 53% of NSCLC cases exhibit mutations in three genes: the Kirsten rat sarcoma virus (*KRAS*), epidermal growth factor receptor (*EGFR*), or B-Raf Proto-Oncogene, Serine/Threonine Kinase (*BRAF*). In addition, driver gene fusions and splicing variants are found in 10–15% of patients [3,4]. Among the prevalent gene fusions observed in NSCLC, three genes encode membrane receptors, namely, anaplastic lymphoma receptor tyrosine kinase (*ALK*), the ret proto-oncogene (*RET*), and the ROS proto-oncogene 1, receptor tyrosine kinase (*ROS1*). *ALK* fusion has a frequency of 5–7%, while *RET* and *ROS1* fusions have a similar frequency of 1–2% [5]. The fusion of the echinoderm microtubule-associated protein-like 4 (*EML4*) and the kinesin family member 5B (*KIF5B*) are significant fusion partners of the *ALK* gene. These fusions are important biomarkers for predicting responses to tyrosine kinase inhibitors (TKIs). As per the National Comprehensive Cancer Network (NCCN) guidelines, patients with *ALK*-fusion-positive tumors have shown positive responses to ALK TKIs. Alectinib has demonstrated superior efficacy with respect to crizotinib as an initial treatment option for these patients [6]. *RET* rearrangement was first identified in NSCLC in 2012 using a next-generation sequencing assay [7]. The kinase domain of the *RET* gene combines with the N-terminal region of various gene partners to form fusions. This fusion leads to the ligand-independent and continuous activation of RET, which stimulates cell proliferation and enhances cell survival [8]. *KIF5B* is the most prevalent partner in 70–80% of cases, followed by *CCDC6* and *NCOA4* [9]. The FDA has approved two TKIs that target RET specifically, namely, selpercatinib and pralsetinib, for the treatment of advanced RET-positive NSCLC [10]. The *ROS1* gene was first discovered in the 1980s, and its role as a proto-oncogene was identified in brain tumors in 2003 [11]. In lung cancer, two ROS1 fusion transcripts, namely, SLC34A2-ROS1 and CD74-ROS1, were initially identified as proto-oncogenes [12]. The rearrangement of *ROS1* primarily occurs in exons 32, 34, 35, or 36, as well as in introns 31 or 33 [13,14]. Among the fusion partners, *CD74* is the most prevalent (accounting for 38–54% of cases), followed by *EZR* (13–24%), *SDC4* (9–13%), *SLC34A2* (5–10%), and *GOPC* (2–3%) [14,15,16]. The test for *ROS1* rearrangement is currently recommended for all cases of metastatic lung carcinomas. The FDA has approved two tyrosine kinase inhibitors, crizotinib and entrectinib, as initial treatment options.

Numerous rare fusions involving receptor tyrosine kinase (RTK) have been detected in lung cancer, including fibroblast growth factor receptor (*FGFR*), neuregulin 1 (*NRG1*), and MET Proto-Oncogene, Receptor Tyrosine Kinase (*MET*). The FGFR family comprises four members: FGFR1, FGFR2, FGFR3, and FGFR4 [17]. Although *FGFR* fusions are infrequent, they have been observed in lung cancer, with the most prevalent being *FGFR3-TACC3* [18]. Drugs targeting FGFRs are currently being developed, such as erdafitinib and ARQ-087, which have shown limited responses with manageable toxicity [19,20]. NRG1, a ligand of the Human Epidermal Growth Factor Receptor (HER) family, stimulates the activation of RTKs. *NRG1* fusion is very rare, and drug development is supported by small-cohort studies and case reports [21,22]. *MET* fusion (*TPR-MET*) was initially discovered in an osteogenic sarcoma cell line, with an incidence of approximately 0.5% in lung cancer. Despite the identification of numerous fusion partners, the biological and therapeutic implications remain to be evaluated [23].

Various diagnostic techniques can be used to identify gene fusions, including immunohistochemistry (IHC), fluorescence in situ hybridization (FISH), reverse transcription quantitative polymerase chain reaction (RT-qPCR), and next-generation sequencing (NGS). FISH has traditionally been considered the gold standard for detecting RTK fusions induced by chromosomal translocations and intrachromosomal rearrangements [24]. RTK gene fusions often lead to elevated mRNA and protein levels, making RT-qPCR and IHC effective alternatives [25]. A positive ALK IHC result has been utilized to prescribe ALK inhibitors [24]. FISH requires no sophisticated equipment and is valuable as a validation tool after positive IHC or NGS results [26]. Despite being considered as a benchmark, FISH is associated with higher costs, technical challenges related to limited tumor cell availability, and high operator variability [27,28]. Additionally, it examines only one alteration per test, necessitating more material for comprehensive analyses, and has limitations in detecting small intrachromosomal rearrangements due to its analytical resolution of 100–200 kb, limiting sensitivity [29]. IHC, on the other hand, is more cost-effective than FISH and is useful for preselecting tumors for confirmatory FISH testing, offering excellent sensitivity with specific antibodies. However, it does not identify the fusion partner and relies on antibody specificity [30]. With the growing demand for comprehensive genomic assessment, NGS-based fusion detection is becoming a preferred approach. NGS provides extensive genetic information, including fusion breakpoints at single-nucleotide resolution and the detection of unknown fusion partners. Despite its numerous advantages, NGS also has many limitations, such as restricted access to panels, dependency on sample quality and quantity, the need for bioinformatic support increasing overall costs, and the requirement for complex and expensive equipment [31]. RT-qPCR-based methods offer high specificity, providing robust and detailed information. Using ROS1 fusion-specific primers, RT-qPCR demonstrates outstanding performance with 100% sensitivity and 85.1% specificity for ROS1 fusion detection [32]. However, its results are contingent on RNA quality, and primers must be designed based on known fusion partners, as it is unable to detect unknown partners.

Initially, identifying and characterizing gene fusions in clinical biopsies focused on single alterations using the polymerase chain reaction (PCR). However, the continuously expanding array of targetable fusions prompted the advancement of multiplex techniques. These methods enable the investigation of multiple alterations in a single assay.

Xeno nucleic acid (XNA) is a synthetic DNA analog, originally developed to store genetic information and evolve in response to external stimuli [33]. XNAs have proven to be highly effective in binding to specific regular DNA sequences, making them useful as molecular clamps in quantitative real-time PCR or as precise molecular probes for identifying specific nucleic acid sequences [34]. The attachment of XNA to its designated sequence blocks the extension of the DNA strand by DNA polymerase. In cases where there is a mismatch in the target site sequence, the XNA–DNA duplex lacks stability, allowing the extension of the strand by DNA polymerase [35,36].

In this study, we present the development and verification of a novel multiplex RT-qPCR assay that uses XNA. The assay is designed to identify gene fusions that are frequently observed in lung cancer cases in a simultaneous and qualitative manner. The fusion biomarker assay can detect fusion events in *ALK*, *RET*, and *ROS1* target genes.

## 2. Materials and Methods

### 2.1. Clinical Samples and Cell Lines

Twenty preserved samples in formalin-fixed paraffin-embedded (FFPE) format were provided by the lung External Quality Assessment (EQA) program conducted by the European Society of Pathology (ESP). Details about the EQA scheme can be found at http://lung.eqascheme.org/info/public/alk/index.xhtml (accessed on 19 February 2024). The ESP schemes followed the required guidelines for EQA programs in the field of molecular pathology [37]. The clinical information was not provided by the organization. Fifty-seven de-identified FFPE samples from lung cancer patients were obtained from Amsbio (Cambridge, MA, USA). Approval for ethical considerations was granted by the institutional review board (IRB) at WIRB-Copernicus Group, Inc., and sample collection was conducted with written informed consent. Specimen details are listed in Appendix A. Fusion-positive and -negative FFPE reference materials were obtained from Horizon Discovery (Cambridge, UK). RNAs were extracted from FFPE sections using the RNeasy FFPE kit (QAIGEN, Hilden, Germany) according to the manufacturer’s instructions. Two lung cancer cell lines, A549 (a fusion-negative cell line) and H2228 (an EML4-ALK V3a/b-positive cell line) were purchased from the American Type Culture Collection (ATCC). A CCDC6-RET-positive cell line, LC-2/ad, was purchased from Millipore Sigma (Burlington, MA, USA). An SLC34A2-ROS1-positive cell line, HCC78, was purchased from the German Collection of Microorganisms and Cell Cultures (DSMZ). Total RNA was isolated from 1 million cells using a Direct-zol RNA miniprep kit (Zymo Research, Irvine, CA, USA) according to the manufacturer’s instructions. Total lung tissue RNA was purchased from Biochain (Newark, CA, USA). RNAs were quantified using the Qubit RNA BR assay kit (Invitrogen, Waltham, MA, USA).

### 2.2. Detection of ALK, RET, or ROS1 Fusion Using RT-qPCR

*ALK*, *RET*, or *ROS1* fusion was detected using the Qfusion^TM^ ALK fusion detection kit, the Qfusion^TM^ RET fusion detection kit, and the Qfusion^TM^ ROS1 fusion detection kit (DiaCarta Inc., Pleasanton, CA, USA), which can simultaneously detect thirteen *ALK* fusions, seven *RET* fusions, and seven *ROS1* fusions, respectively (Appendix A). Briefly, 50 ng of RNAs was subjected to RT-qPCR using Bio-Rad CFX384 (Bio-Rad Laboratories, Inc., Hercules, CA, USA). The thermal cycling condition was 10 min at 50 °C for reverse transcription (RT) and 2 min at 95 °C for RT inactivation, followed by 45 cycles of 5 s at 95 °C for denaturation, 40 s at 70 °C for XNA binding, 30 s at 64 °C for primer binding, and elongation with a slow ramp rate (3 °C per second). The experimental workflow is shown in Appendix A.

### 2.3. Determination of Analytical Sensitivity

The analytical sensitivity was assessed using synthetic double-stranded DNAs (Appendix A), which were purchased from Integrated DNA Technologies (Coralville, IA, USA) as gBlocks gene fragments. Three distinct copy numbers (500, 100, and 50 copies) of gBlocks were used as inputs, and the RT-qPCR assay was conducted with 18 replicates.

To determine the limit of detection of RNA fusion targets in cell lines, 50 ng of total RNA was used as input. The total RNA from each cell line underwent serial dilution with normal lung tissue RNA (Biochain) and then was checked using the Qfusion^TM^ ALK fusion detection kit, the Qfusion^TM^ RET fusion detection kit, or the Qfusion^TM^ ROS1 fusion detection kit (DiaCarta Inc.). Similarly, for RNA fusion targets in the FFPE RNA reference standard (Horizon Discovery), RNAs were diluted with normal lung tissue FFPE RNA (Horizon Discovery) and tested using the Qfusion^TM^ assays.

### 2.4. Detection of ALK, RET, or ROS1 Fusion Using Next-Generation Sequencing (NGS)

Lung cancer-specific fusions were analyzed using an OptiSeq^TM^ lung cancer fusion NGS panel (DiaCarta Inc., Pleasanton, CA, USA). Fifty nanograms of RNAs was used as an input for library preparation. The libraries were sequenced on an Illumina MiSeq platform (Illumina, San Diego, CA, USA) with read lengths of 150 bases, generating approximately 0.1 million paired end reads per library. The obtained results were analyzed using the QIAGEN CLC Genomics Workbench version 20.0.4 (QIAGEN, Aarhus, Denmark) for fusion detection. A fusion event was deemed confirmed if a minimum of 10 reads were identified as crossing the fusion junction.

### 2.5. Preparation of Fusion and Wild-Type Transcripts by In Vitro Transcription

The EML4-ALK V1 and wild-type ALK transcripts were synthesized using the Megascript T7 transcription kit (Thermo Fisher Scientific, Carlsbad, CA, USA) following the manufacturer’s guidelines. The RNAs were then purified using the RNA Clean & Concentrator-5 kit (Zymo Research) and quantified using the Qubit RNA BR assay kit (Invitrogen), following the manufacturer’s guidelines.

## 3. Results

### 3.1. Analytical Sensitivity, Limit of Detection (LoD)

We conducted 18 replicate reactions to check the analytical sensitivity of each Qfusion^TM^ assay designed for detecting *ALK*, *RET*, or *ROS1* fusions. We used 500, 100, and 50 copies of synthetic double-stranded fusion templates for the test (Appendix A). The analytical sensitivity was expressed as the number of copies detected per reaction. The results demonstrated a 100% detection rate when testing with 500, 100, and 50 copies per reaction for most templates. However, there was a slightly lower detection rate of 89% for the EML4-ALK V6 template and 83% for the GOPC-ROS1 G8;R35 template (Table 1). Therefore, the limit of detection for all three detection assays was established at 50 copies of the target per reaction.

### 3.2. Enhancement of Qfusion^TM^ Assay Specificity and Sensitivity with XNA

Off-target amplification is a common issue in multiplex PCR-based assays, especially when there is a significant background of wild-type sequences [38]. This often leads to false positives when detecting mutations or fusions due to the promiscuous binding of primers and probes to sequences that closely resemble the target [35].

To address the issue of non-specific amplification in the presence of abundant wild-type backgrounds, we utilized XNA clamp probes designed to hinder the amplification of wild-type transcripts. XNA forms stable complexes with its targets, while mismatches in fusion partners result in weak complexes, allowing the amplification of fusion targets (Appendix A). Given the characteristics of XNA, we hypothesized that its application could enhance the specificity and sensitivity of the Qfusion^TM^ assays.

First, to evaluate the impact of XNA on specificity, we tested the detection efficiency of each Qfusion^TM^ assay by using 10-fold serial dilutions of ALK, RET, or ROS1 wild-type RNA transcripts in the presence or absence of XNA. As shown in Figure 1a, each assay exhibited non-specific detection of wild-type RNA transcripts. However, the presence of XNA efficiently reduced the detection of wild-type RNAs. This suggests that XNA enhances the assay’s specificity, reducing the occurrence of false positives.

We conducted multiple measurements using varying quantities of EML4-ALK V1 synthetic DNA fragments to evaluate the impact of XNA on assay sensitivity. We performed the tests in the presence and absence of 10,000,000 copies of wild-type ALK DNA fragments. Since the presence of abundant wild types could lead to non-specific amplification in the reaction (Figure 1a), we hypothesized that XNA would enhance the detection limit of the fusion target by preventing non-specific amplification of wild types. Furthermore, we postulated that XNA would not affect the detection of fusion transcripts in the absence of wild-type ALK.

As anticipated, our results affirmed that XNA had no significant impact on the detection of EML4-ALK V1 fusion (Appendix A). In the absence of XNA, the quantification cycle (Cq) value remained unchanged for different copy numbers of fusion targets due to preferential amplification of abundant wild types, masking the amplification of fusion targets (Figure 1b). Therefore, the assay was unable to distinguish between 50 copies and 2.5 copies, resulting in a 50-copy LoD. However, XNA effectively prevented the amplification of wild types, enabling the detection of low copy numbers of fusion targets in a linear, dose-dependent manner ranging from 2.5 to 50 copies. This implies that the LoD was enhanced to the 2.5-copy level, demonstrating increased sensitivity.

In summary, XNA inhibited the amplification of wild types, increasing the specificity and sensitivity of the Qfusion^TM^ assay.

### 3.3. Evaluation of the Qfusion^TM^ Assays’ Performance with Cell Lines and FFPE RNA Reference Standards

The sensitivity and specificity of individual Qfusion^TM^ assays were subsequently assessed using fusion-negative and -positive cancer cell lines. RNA extraction was performed on four cell lines: A549 (fusion-negative), H2228 (EML4-ALK V3/b-positive), LC-2/ad (CCDC6-RET C1; R12-positive), and HCC78 (SLC34A2-ROS1 S4; R32-positive). RNA dilution was conducted using normal lung tissue RNA along with RNAs from cancer cell lines, ranging from 100% to 0% representation of tumor portions. Through the systematic dilution of cancer cell line RNAs, the sensitivity and specificity of the Qfusion^TM^ assay were assessed. The individual XNA-mediated ALK, RET, or ROS1 Qfusion^TM^ assays successfully identified all fusion targets with 1% minimum dilution. The corresponding Cq values for ALK, RET, and ROS1 fusions were 33.62 ± 0.50, 27.03 ± 0.13, and 25.42 ± 0.10, respectively. No fusion signals were detected in the A549 fusion-negative cell line (Figure 2a).

The current gold standard diagnostic method for detecting *ALK*, *RET*, or *ROS1* fusion is FISH. A positive fusion result is typically determined if at least 15% of tumor cells exhibit rearrangement [39]. The Qfusion^TM^ assay is specifically designed to detect fusions in both archived and fresh tissue samples. The Cq values for each Qfusion^TM^ assay at the 15% tumor cell threshold were determined. To ascertain these Cq values, RNAs were prepared from commercially available FFPE reference standards and normal lung FFPE samples. The Qfusion^TM^ assay was then used to analyze various concentrations of reference-standard FFPE RNAs that were diluted with normal lung FFPE RNAs. The resulting Cq values for the Qfusion^TM^ ALK, RET, or ROS1 fusion assays at a 15% tumor fraction were determined as 32.45 ± 0.17, 28.84 ± 0.25, and 25.27 ± 0.03, respectively (Figure 2b,c).

### 3.4. Evaluation of the Qfusion^TM^ Assays’ Performance Using Clinical Samples

In our previous study, we analyzed twenty FFPE patient samples acquired from the EQA program of the ESP using the OptiSeq^TM^ lung cancer fusion NGS panel [40]. This panel, which was specifically designed for the detection of 63 known lung cancer-specific fusion genes, including *ALK*, *RET*, and *ROS1* fusion genes, successfully identified SLC45A-ROS1 fusions in two patient samples (Figure 3a). However, there were no instances of ALK or RET fusions. Subsequently, the same twenty samples were assessed using the Qfusion^TM^ ALK, RET, or ROS1 fusion detection assay. As shown in Figure 3b, the Qfusion^TM^ ROS1 fusion detection assay detected the same patients as ROS1 fusion positive. Conversely, both the Qfusion^TM^ ALK and RET fusion detection assays showed that all tested samples were negative for *ALK* or *RET* fusion.

The consistent findings observed between the Qfusion^TM^ assays and the NGS assay prompted us to extend our investigation to a larger set of patient samples. The Qfusion^TM^ ALK, RET, and ROS1 fusion detection assays were employed to assess 57 distinct clinical FFPE samples obtained from Amsbio (Cambridge, MA, USA). Out of the 57 samples, one patient was identified as *ALK* fusion-positive, one as *RET* fusion-positive, and one as *ROS1* fusion-positive (Figure 4a). Interestingly, one patient sample, AMS029, exhibited concurrent *ALK* and *RET* fusions. To validate the outcomes derived from the Qfusion^TM^ assay and determine the specific fusion genes implicated, all fusion-positive samples underwent a reanalysis employing individual PCR amplicons. The Qfusion^TM^ assay product (amplicon) served as the template for the second individual PCR, wherein a single primer pair and probe were employed to discern the specific fusion variant.

The results of the singleplex individual PCR test showed that the AMS029 patient harbored two types of EML4-ALK fusions, V1 and V7 variants, and one CCDC6-RET fusion. In the case of the ROS1 fusion-positive AMS056 patient, the singleplex individual PCR identified the CD74-ROS1 fusion variant (Figure 4b). Subsequently, each individual PCR product was analyzed through sanger sequencing, which confirmed the presence of the corresponding fusion variants (Appendix A).

## 4. Discussion

Chromosomal translocations resulting in the formation of fusion genes represent a significant causative factor in cancer. Accurate diagnosis of these events is crucial for effective treatment. The commonly used diagnostic method, FISH, relies on pre-existing annotations, exhibiting low throughput and limited resolution. In contrast to FISH, RNA NGS sequencing (RNAseq) provides a high-resolution approach to detect fusion genes capable of identifying both established and novel fusions. While the high-throughput nature of RNAseq expands diagnostic possibilities, it also poses a challenge in terms of increased false-positive rates. This indicates potential inaccuracies in fusion gene calls and the identification of non-driver fusion events, such as recurrent chimeric fusion RNAs in non-cancerous tissues and cells [41].

Incorporating RNAseq into routine clinical practice for primary fusion detection is impractical due to its elevated cost, long turnaround time, and complex procedural and data analysis requirements. Therefore, our focus has shifted towards developing a cost-effective, high-throughput fusion detection assay utilizing a multiplex-based reverse RT-qPCR approach.

Numerous RT-qPCR assays have been published for the detection of *ALK*, *RET*, and *ROS1* fusion genes. Kuang et al. conducted an analysis of RT-qPCR and NGS for *EML4-ALK* detection, reporting a concordance rate of 95.16%. In approximately 5% of discordant samples, RT-qPCR further identified additional ALK fusions among those with negative NGS results [42]. Lu et al. reported the incidence of EML4-ALK detected by IHC (9.51%), RT-qPCR (11.62%), and NGS (5.84%) [43], suggesting that RT-qPCR has the potential advantage in clinical scenarios for ALK fusion detection. Regarding *RET* fusions, Kim et al. investigated the frequency of the *KIF5B-RET* fusion gene and its association with other oncogenic drivers. They analyzed expression in archival tumor tissues using RT-qPCR, detecting the fusion gene in 5.8% of cases, which was confirmed by FISH [44]. In another study, Tsai et al. aimed to detect two types of *RET* fusions, *KIF5B-RET* and *CCDC6-RET*, simultaneously, using a multiplex RT-qPCR platform. They identified *RET* rearrangements in 2.5% of patients, suggesting the feasibility of screening for *RET* fusions in a large population [45]. Another large-scale study demonstrated the ease of *ROS1* fusion detection by RT-qPCR using a commercial *ROS1* fusion detection RT-qPCR kit. Zhang et al. analyzed 2358 cytological specimens and identified *ROS1* fusions in 41 patients (1.95%). Furthermore, they monitored 14 patients treated with crizotinib but found no significant differences in the objective response rate, highlighting the prognostic implications of RT-qPCR based fusion detection [46].

Digital PCR (dPCR) is emerging as a promising alternative to NGS for cancer biomarker testing due to its straightforward workflow, minimal sample requirements, rapid turnaround time, and cost-effectiveness [47]. However, conventional dPCR’s clinical utility is hindered by its intrinsic limitation in multiplexing, which restricts the simultaneous assessment of all actionable biomarkers in a single assay with a limited sample quantity. In an attempt to address this limitation, Cabrera et al. developed a multiplex dPCR panel for fusion detection. Their study evaluated the analytical performance and concordance of this multiplex dPCR panel with NGS, achieving a concordance rate exceeding 97%. This demonstrates that multiplex dPCR significantly reduces turnaround time and maintains analytical sensitivity and reactivity comparable to currently accepted amplicon-based NGS methodologies. The simplified workflow and analysis of dPCR offer a promising solution for enhancing accessibility to biomarker testing [48].

To design assays for the identification of gene fusion transcripts, we used our existing XNA technology, which was previously used for detecting single-nucleotide polymorphism (SNP) in genomic DNA samples [35]. Our panels targeted 28 well-known fusion transcripts, encompassing over 90% of ALK, RET, or ROS1 fusion transcripts [49]. We systematically tested the assays on in vitro transcribed RNAs, cell lines, and FFPE samples and successfully identified both known fusion transcripts and fusions in patient FFPE samples with previously unknown fusion status.

The use of a multiplex PCR method can result in decreased specificity and sensitivity compared to singleplex PCR [50,51]. As illustrated in Figure 1, non-specific amplification of wild-type ALK transcripts was observed despite using fusion-specific primer and probe sets. However, the introduction of XNA effectively mitigated this non-specific amplification, thereby increasing assay specificity and sensitivity. Our utilization of XNA technology, originally employed for SNP detection on DNA, has been extended to include fusion detection on RNA.

All three Qfusion^TM^ assays were able to detect synthetic targets even at levels as low as 50 copies per reaction, with 18 replicate measurements. In addition, these assays were able to identify corresponding targets in cell lines and FFPE RNA references even when the tumor fraction was as low as 1%. However, it is noteworthy that the criteria for fusion positivity was established based on FISH standards, where a tumor cell population of 15% exhibiting fusion is considered a positive case.

When applying this criterion to 57 FFPE samples from patients with unknown fusion, we found two fusion-positive cases, each exhibiting a frequency of 1.8% for ALK fusion, RET fusion, and ROS1 fusion. Interestingly, one patient displayed two distinct fusions, ALK and RET fusions, which was confirmed by singleplex PCR. This is a rare co-occurrence of such dual fusions in a single patient and could be due to the nature of intratumor heterogeneity [52]. The low frequency of fusion-positive cases among lung cancer patients observed in this study indicates the challenges posed by the relative rarity of these events in the broader patient population. This underscores the importance of using sensitive and comprehensive diagnostic approaches to detect such infrequent but clinically significant occurrences, which can contribute to our understanding of the prevalence of fusion-positive cases in lung cancer.

Patients harboring RTK fusions have exhibited favorable responses to FDA-approved TKIs. Specifically, alectinib and crizotinib have demonstrated efficacy for ALK fusions [6], while selpercatinib and pralsetinib have shown effectiveness for RET fusions [10]. Additionally, crizotinib and entrectinib have been effective for ROS1 fusions [46]. For instance, a cohort of fifty-three patients received crizotinib and were followed for an average duration of 62.6 months. The objective response rate (ORR) was 72%, comprising 6 complete responses and 32 partial responses. This investigation underscores the efficacy of crizotinib in treating ROS1 fusion-positive patients [46]. In line with our assay development objectives, we aim to not only facilitate disease diagnosis but also monitor treatment response. Our goal is to extend the applicability of our assays to longitudinal studies, enabling the evaluation of treatment response over time.

The timely and accurate identification of biomarkers that can be acted upon is a significant step forward in making testing more accessible. To achieve this, we are introducing RT-qPCR panels that comprehensively cover actionable targets and show a high level of agreement with NGS results. These panels have a streamlined workflow and simple analysis, making them a promising solution to improve the accessibility of biomarker testing, thereby contributing to the advancement of personalized medicine guidance.

## Figures and Tables

**Figure 1 diagnostics-14-00488-f001:**
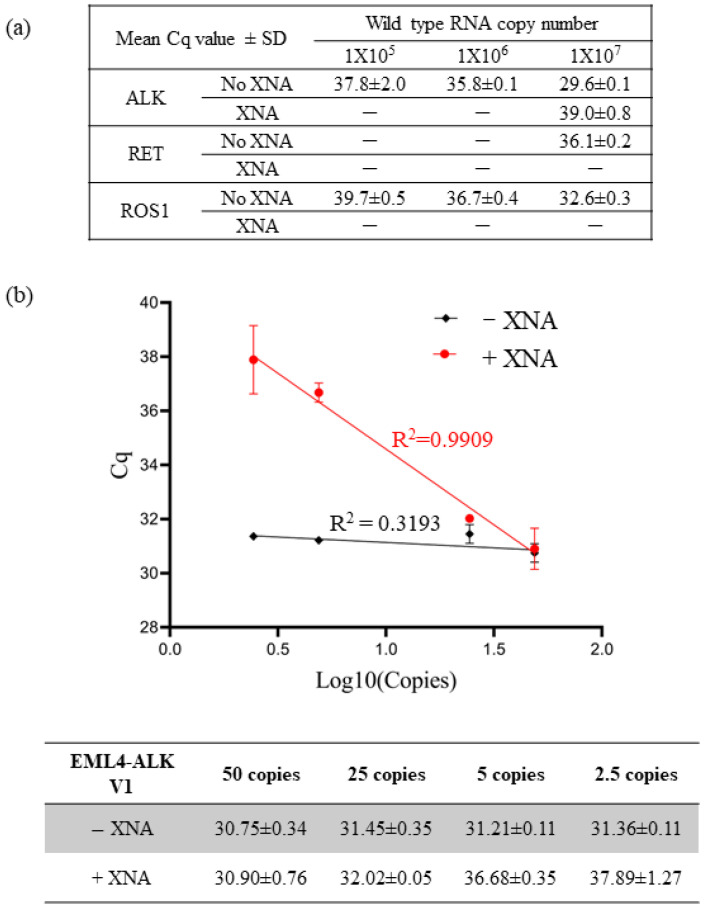
Enhancing assay specificity and sensitivity with XNA. (**a**) XNA enhanced specificity by preventing non-specific amplification of abundant wild-type sequences. (**b**) XNA improved assay sensitivity by inhibiting the wild-type backgrounds.

**Figure 2 diagnostics-14-00488-f002:**
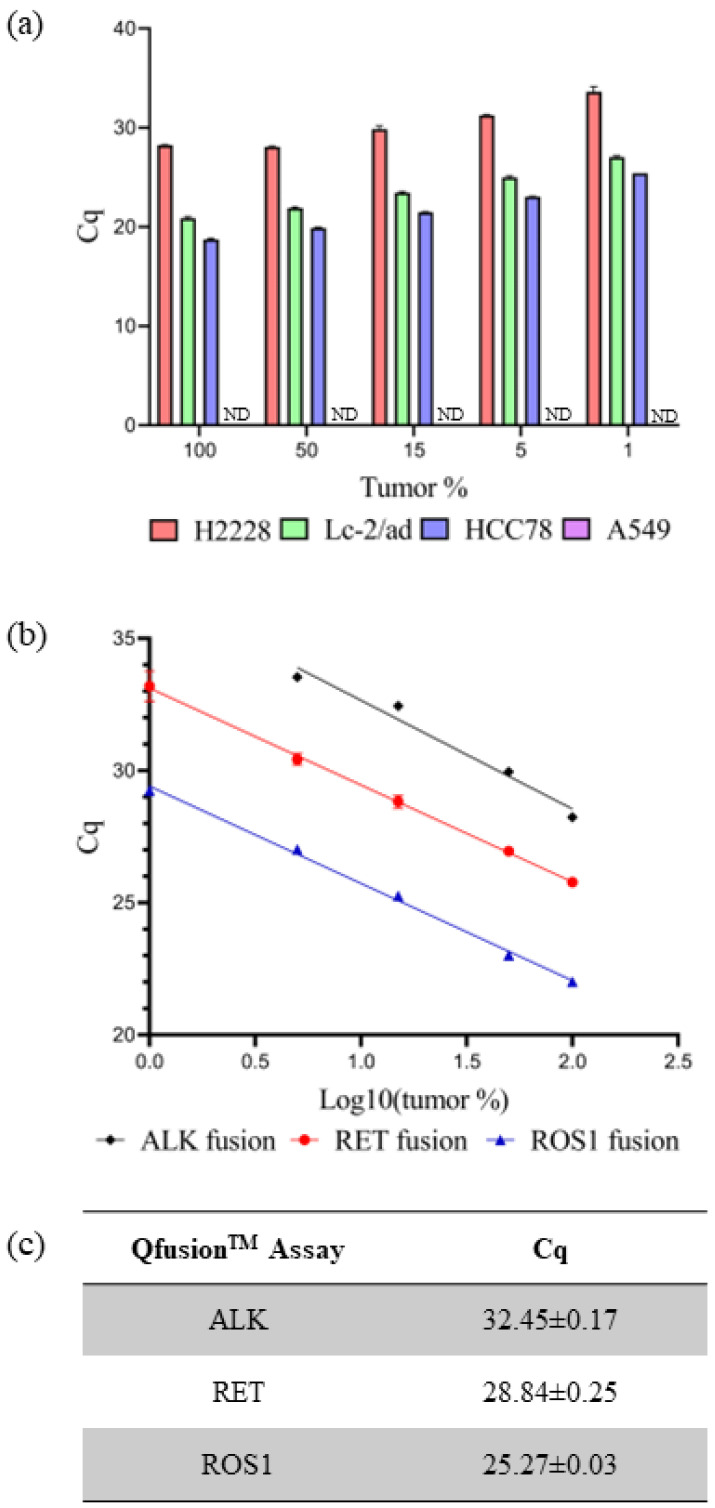
Evaluating Qfusion^TM^ assay performance. (**a**) Each Qfusion^TM^ assay was tested with the corresponding cell lines: A549, H2228 (EML4-ALK), LC-2/ad (CCDC6-RET), and HCC78 (SLC45A-ROS1). The A549 was a fusion-negative cell line and Cq was not determined (ND). (**b**,**c**) FFPE RNA reference standard was diluted with normal FFPE RNA and analyzed with each Qfusion^TM^ assay. The x-axis represents the logarithm base 10 value of tumor percentage. Cq values were determined for 15% of tumor samples.

**Figure 3 diagnostics-14-00488-f003:**
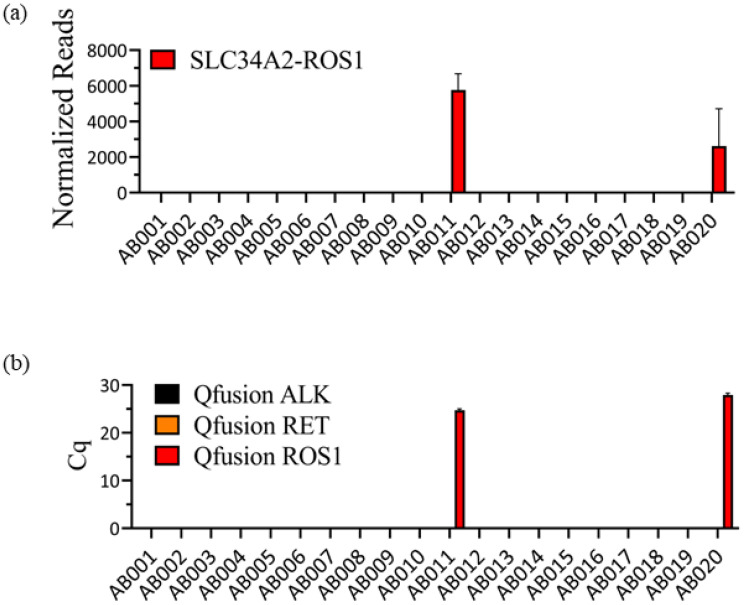
Consistency between OptiSeq^TM^ lung cancer fusion NGS panel and Qfusion^TM^ ALK, RET, or ROS1 fusion detection assay results. (**a**) Analysis of twenty FFPE samples using the OptiSeq^TM^ lung cancer fusion NGS panel revealed the presence of SLC34A2-ROS1 fusion in two patient samples. The Y-axis represents normalized reads, and the error bars indicate the standard deviations of three replicates. (**b**) The same twenty FFPE samples were subjected to analysis using Qfusion^TM^ assays. The results confirmed ROS1 fusion positivity in two patients with SLC34A2-ROS1 fusion. Qfusion^TM^ ALK and RET fusion detection assays showed no ALK and RET fusions. The error bars depict the standard deviations of three replicates.

**Figure 4 diagnostics-14-00488-f004:**
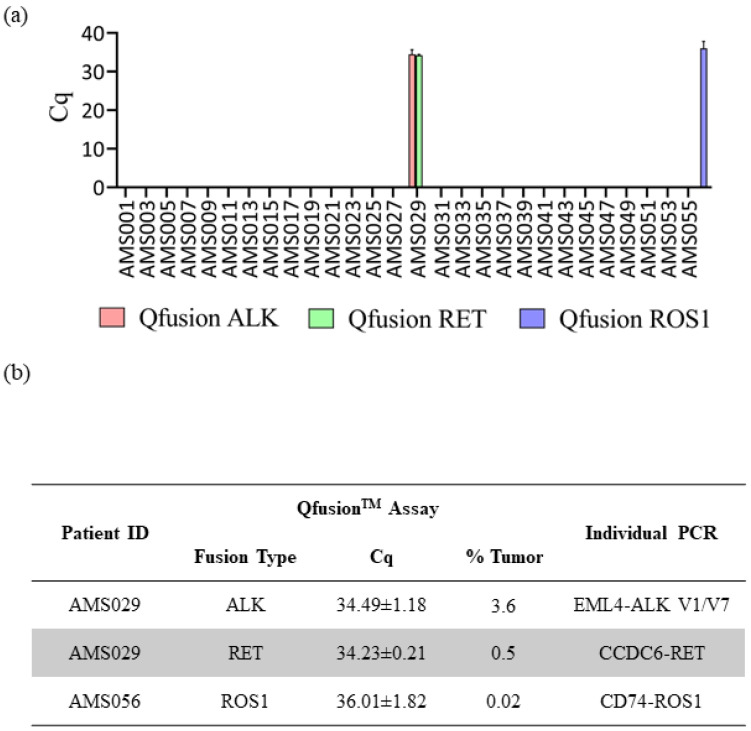
Assessment of Qfusion^TM^ assays’ performance utilizing clinical samples. (**a**) Results from the analysis of 57 FFPE lung cancer patient samples using the Qfusion^TM^ ALK, RET, or ROS1 fusion detection assay. The data reveals the identification of one patient with ALK and RET fusions and one patient with ROS1 fusion. (**b**) Validation of distinct fusion events through singleplex RT-qPCR. The identified ALK fusions included EML4-ALK V1/V7, while the RET fusion involved CCDC6-RET. Additionally, a single ROS1 fusion event, CD74-ROS1, was confirmed.

**Table 1 diagnostics-14-00488-t001:** (**a**) Analytical sensitivity of the Qfusion^TM^ ALK fusion detection assay. (**b**) Analytical sensitivity of the Qfusion^TM^ RET fusion detection assay. (**c**) Analytical sensitivity of the Qfusion^TM^ ROS1 fusion detection assay.

(**a**)													
**Copies ***	**V1**	**V2**	**V3a**	**V3b**	**V5a**	**V5b**	**V6**	**V7**	**V8a**	**V8b**	**K15**	**K17**	**K24**
**500**	100	100	100	100	100	100	100	100	100	100	100	100	100
**100**	100	100	100	100	100	100	100	100	100	100	100	100	100
**50**	100	100	100	100	100	100	89	100	100	100	100	100	100
(**b**)													
**Copies ***	**CCDC6-RET** **C1;R12**		**CCDC6-RET** **C2;R12**		**KIF5B-RET** **K15;R12**		**KIF5B-RET** **K16;R12**		**KIF5B-RET** **K22;R12**		**KIF5B-RET** **K23;R12**	**NCOA4-RET** **N8;R12**	
**500**	100		100		100		100		100		100	100	
**100**	100		100		100		100		100		100	100	
**50**	100		100		100		100		100		100	100	
(**c**)													
**Copies ***	**SDS4-ROS1** **S2;R32**		**SLC34A2-ROS1** **S4;R32**		**SLC34A2-ROS1** **S13;R32**		**CD74-ROS1** **C6;R34**		**EZR-ROS1** **E10;R34**		**GOPC-ROS1** **G8;R35**	**GOPC-ROS1** **G4;R36**	
**500**	100		100		100		100		100		100	100	
**100**	100		100		100		100		100		100	100	
**50**	100		100		100		100		100		83	100	

* Amount of input per reaction.

## Data Availability

The data presented in this study are available within the article.

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
