# Peer review of "A Highly Sensitive XNA-Based RT-qPCR Assay for the Identification of ALK, RET, and ROS1 Fusions in Lung Cancer"

_diagnostics, 2024, doi:10.3390/diagnostics14050488_

Round 1

Reviewer 1 Report

Comments and Suggestions for Authors

The article presents a multiplexed RT-qPCR assay for the detection of lung cancer gene fusions, specifically ALK, RET, and ROS1. The assay is based on xenonucleic acid (XNA) molecular clamping technology and designed to quantitatively detect these gene fusions in formalin-fixed paraffin-embedded (FFPE) samples. The manuscript is well-written; however, I have some concerns below which may help to strengthen the conclusions and improve the scientific rigor.

Major comments

1. While the assay shows promising results in identifying lung cancer gene fusions, it would be helpful to compare its performance with other established methods for fusion gene detection, such as immunohistochemistry (IHC) or fluorescence in situ hybridization (FISH). This would provide a more comprehensive understanding of the assay's advantages and limitations.

2. It would be interesting to explore the clinical implications of the assay's findings. Do the identified gene fusions predict response to tyrosine kinase inhibitors? Are there any prognostic implications? 

3. It would be beneficial to expand the validation of the assay to a larger and more diverse set of clinical samples, including samples from different ethnic groups and representing various stages of lung cancer. 

4. The assay's adaptability to cell-free RNAs is a promising feature that could be further explored. It would be interesting to compare the performance of the assay on cell-free RNAs with that on FFPE samples to assess its potential for liquid biopsies or other non-invasive diagnostic methods.

Minor comments:

5. It would be beneficial to include a flowchart or diagram to illustrate the experimental workflow.

6. The introduction section could provide a more comprehensive background on the importance of gene fusions in lung cancer and the current challenges in their detection.

7. It would be helpful to provide a more focused hypothesis regarding the use of XNA in the multiplex RT-qPCR assay.

8. Provide a more comprehensive description of the multiplex RT-qPCR assay, including the primers used and their specificities.

9. Proof-reading for the language is needed, including the grammatical errors and some spelling errors.

Comments on the Quality of English Language

Proof-reading for the language is needed, including the grammatical errors and some spelling errors.

Author Response

Dear Reviewer,

We would like to express our gratitude for your thoughtful review of our manuscript titled “A Highly Sensitive XNA-based RT-qPCR Assay for the Identification of ALK, RET, and ROS1 Fusions in Lung Cancer" submitted for consideration to Diagnostics. Your detailed feedback and constructive criticism are invaluable in improving the quality and clarity of our work. We appreciate the time and effort you have dedicated to evaluating our manuscript and providing insightful comments.

In response to your critiques and suggestions, we have carefully revisited our work and made revisions aimed at addressing each point raised. In this rebuttal letter, we provide a point-by-point response to your comments, outlining the changes made and offering explanations where necessary. We hope that our responses adequately address your concerns and demonstrate our commitment to producing high-quality research.

Once again, we thank you for your valuable feedback and the opportunity to improve our manuscript. We look forward to your reevaluation of the revised version and the possibility of its eventual publication in Diagnostics.

Sincerely,

Comments and Suggestions for Authors

The article presents a multiplexed RT-qPCR assay for the detection of lung cancer gene fusions, specifically ALK, RET, and ROS1. The assay is based on xenonucleic acid (XNA) molecular clamping technology and designed to quantitatively detect these gene fusions in formalin-fixed paraffin-embedded (FFPE) samples. The manuscript is well-written; however, I have some concerns below which may help to strengthen the conclusions and improve the scientific rigor.

Major comments

  1. While the assay shows promising results in identifying lung cancer gene fusions, it would be helpful to compare its performance with other established methods for fusion gene detection, such as immunohistochemistry (IHC) or fluorescence in situ hybridization (FISH). This would provide a more comprehensive understanding of the assay's advantages and limitations.

Thank you for your valuable comments and inquiries.

We fully acknowledge and appreciate your suggestion regarding the inclusion of a performance comparison of RT-qPCR with IHC or FISH. Indeed, supplementing our study with comparisons to these established methods would provide a more comprehensive understanding of the efficacy of our assays. Unfortunately, we are unable to conduct these experiments at present due to constraints such as our current budgetary limitations and the unavailability of necessary equipment, Nonetheless, we will explore avenues to address these aspects in near future studies as resources permit.

  1. It would be interesting to explore the clinical implications of the assay's findings. Do the identified gene fusions predict response to tyrosine kinase inhibitors? Are there any prognostic implications?

Indeed, the identified gene fusions, such as ALK, RET, or ROS1, are associated with receptor tyrosine kinases, and their fusion with partner genes results in constitutive activation or overexpression in cancer. several FDA-approved tyrosine kinase inhibitors (TKIs) effectively counteract the oncogenic effects of these fusions. Monitoring drug response or/and Minimal Residual Disease (MRD) using our assays is a primary objective of our assay development efforts. We aim to broaden the applicability of our assays for longitudinal studies, enabling the evaluation of treatment response over time.  As per your suggestion, we have included additional discussion regarding the clinical implications of our assays (Line 388 – 398).

  1. It would be beneficial to expand the validation of the assay to a larger and more diverse set of clinical samples, including samples from different ethnic groups and representing various stages of lung cancer. 

We sincerely appreciate your suggestion. As a proof of concept, we analyzed seventy-seven FFPE samples in this study. However, we acknowledge your valuable point regarding the need to expand both the sample size and criteria in future studies. We recognize the importance of expanding the sample scope for a more robust analysis and testing for different ethnic groups. We plan to address this in future research endeavors.

  1. The assay's adaptability to cell-free RNAs is a promising feature that could be further explored. It would be interesting to compare the performance of the assay on cell-free RNAs with that on FFPE samples to assess its potential for liquid biopsies or other non-invasive diagnostic methods.

We appreciate your continued input and suggestions. Recognizing the challenges associated with obtaining lung tissue samples, we acknowledge the increasing importance of liquid biopsies as a viable alternative. While our assays have not yet been tested with cell-free RNAs, we are actively planning to assess their adaptability to cfRNAs in a future large-scale study. We are confident that our assays are well-suited for cfRNA analysis. It is worth noting that another research group has successfully detected EML-ALK fusion using RT-qPCR with a sensitivity of 65% and specificity of 100% in platelets and plasma. [1]. However, due to time and sample availability limitations, we were unable to include cfRNA assay validation in the current proof-of-concept study.

Thank you for bringing this valuable point to our attention, and we look forward to exploring the potential of cfRNA analysis with our assays in future research endeavors.

Minor comments:

  1. It would be beneficial to include a flowchart or diagram to illustrate the experimental workflow.

We wholeheartedly agree with your suggestion to include a diagram illustrating the experimental workflow. As per your recommendation, we have now incorporated the workflow as a new Supplementary Figure S2 in the revised manuscript. Thank you for your valuable input, which has undoubtedly enhanced the clarity and comprehensibility of our research findings.

  1. The introduction section could provide a more comprehensive background on the importance of gene fusions in lung cancer and the current challenges in their detection.

Thanks for your suggestions. We made an effort to enhance the introduction section by providing additional information on gene fusions in lung cancer and the challenges associated with their detection. The inclusion of other oncogenic fusion drivers such as FGFR1, NRG1, and MET in the introduction section enriches the content and provides a broader context for the study. We also outlined the advantages and drawbacks of the four major fusion detection methods: IHC, FISH, NGS, and RT-PCR. The supplementary introduction segments extend from line 65 to line 98 in the revised manuscript.

  1. It would be helpful to provide a more focused hypothesis regarding the use of XNA in the multiplex RT-qPCR assay.

As per your suggestion, concise hypotheses have been included at the beginning of result section 3.2 (Line 203 - 207). Additionally, to enhance understanding of XNA, a new Supplementary Figure S3 has been added, illustrating the working principle of XNA.

  1. Provide a more comprehensive description of the multiplex RT-qPCR assay, including the primers used and their specificities.

As the assay discussed in this study is commercial, details such as primer and probe sequences, as well as the specific type of XNA used, cannot be disclosed legitimately. Thank you for acknowledging this.

  1. Proof-reading for the language is needed, including the grammatical errors and some spelling errors.

Thanks for the suggestions. We have thoroughly reviewed the manuscript for grammatical and spelling errors, and we are confident that it has been significantly improved.

Thank you for your attention to detail and effort in enhancing the quality of the manuscript. We believe that these improvements will make your comments a meaningful contribution to the overall impact of the study.

References

  1. Nilsson, R.J.; Karachaliou, N.; Berenguer, J.; Gimenez-Capitan, A.; Schellen, P.; Teixido, C.; Tannous, J.; Kuiper, J.L.; Drees, E.; Grabowska, M.; et al. Rearranged EML4-ALK fusion transcripts sequester in circulating blood platelets and enable blood-based crizotinib response monitoring in non-small-cell lung cancer. Oncotarget 2016, 7, 1066-1075, doi:10.18632/oncotarget.6279.

Reviewer 2 Report

Comments and Suggestions for Authors

The reviewed manuscript is dedicated to development and validation of a qPCR for detection of ALK, ROS1, RET fusions that are biomarkers for TKI therapy. The presented results are interesting for scientists, specializing on the field of cancer molecular diagnostics. However, several issues could be amended before acceptance.

Major issues:

1.      While the reported qPCR is obviously commercial, and its details like primer sequences or the XNA type cannot be revealed legitimately, a basic principle lying behind its specificity would underline scientific novelty of the presented work.

2.      At Figure 2b, data points for ALK and RET on the Y-axis indicating amplification in samples with a zero percentage of cancer cells. However, as far as I understand, no unspecific signal was presented at other figures, and the presented assay is absolutely specific exploiting XNA probes as blockers. This controversy looks odd and needs to be clarified.

3.      A number of various qPCR assays for detection of ALK, ROS1, RET fusions has been published. In that light, more background information in Discussion would be highly appreciated. Especially interesting is a comparison of their analytical characteristics. Also, not only qPCR, FISH and NGS are suitable for fusions genes testing. For instance, digital PCR is exceptionally sensitive and has been successfully applied for cfDNA testing.

Author Response

Dear Reviewer,

We would like to express our gratitude for your thoughtful review of our manuscript titled “A Highly Sensitive XNA-based RT-qPCR Assay for the Identification of ALK, RET, and ROS1 Fusions in Lung Cancer" submitted for consideration to Diagnostics. Your detailed feedback and constructive criticism are invaluable in improving the quality and clarity of our work. We appreciate the time and effort you have dedicated to evaluating our manuscript and providing insightful comments.

In response to your critiques and suggestions, we have carefully revisited our work and made revisions aimed at addressing each point raised. In this rebuttal letter, we provide a point-by-point response to your comments, outlining the changes made and offering explanations where necessary. We hope that our responses adequately address your concerns and demonstrate our commitment to producing high-quality research.

Once again, we thank you for your valuable feedback and the opportunity to improve our manuscript. We look forward to your reevaluation of the revised version and the possibility of its eventual publication in Diagnostics.

Sincerely,

Comments and Suggestions for Authors

The reviewed manuscript is dedicated to development and validation of a qPCR for detection of ALK, ROS1, RET fusions that are biomarkers for TKI therapy. The presented results are interesting for scientists, specializing on the field of cancer molecular diagnostics. However, several issues could be amended before acceptance.

Major issues:

  1. While the reported qPCR is obviously commercial, and its details like primer sequences or the XNA type cannot be revealed legitimately, a basic principle lying behind its specificity would underline scientific novelty of the presented work.

Thank you for your feedback. Due to the confidential nature of our XNA, we have appropriately focused on providing additional background information on how XNA could enhance specificity and sensitivity. We have expanded the explanation at the beginning of result section 3.2 (Line 203 - 207). Additionally, we have included a new Supplementary Figure S3 in the revised manuscript, illustrating the principle of XNA blocking. These modifications aim to address the reviewer's concerns and improve the clarity and understanding of our study findings.

  1. At Figure 2b, data points for ALKand RET on the Y-axis indicating amplification in samples with a zero percentage of cancer cells. However, as far as I understand, no unspecific signal was presented at other figures, and the presented assay is absolutely specific exploiting XNA probes as blockers. This controversy looks odd and needs to be clarified.

Indeed, you are correct. A zero percent tumor indicates normal tissue, where no fusion genes would be expected to be found. However, there appears to be a misunderstanding regarding Figure 2b. In Figure 2b, the x-axis represents the log10 value of tumor percentage. Therefore, a value of zero on the x-axis corresponds to a 1% tumor fraction. We analyzed tumor fractions ranging from 1% to 100%, corresponding to values from 0 to 2 on the log10 scale of tumor percentage. To avoid any potential confusion, we have included the following sentence in the figure legend: "The x-axis represents the logarithm base 10 value of tumor percentage."

  1. A number of various qPCR assays for detection of ALK, ROS1, RETfusions has been published. In that light, more background information in Discussion would be highly appreciated. Especially interesting is a comparison of their analytical characteristics. Also, not only qPCR, FISH and NGS are suitable for fusions genes testing. For instance, digital PCR is exceptionally sensitive and has been successfully applied for cfDNA testing.

We appreciate your suggestions. As a response to your suggestions, we have now added a discussion on the previously published RT-qPCR assays (Line 329 – 354) in the discussion section of the revised manuscript. We cited the most representative works for each ALK, RET, and ROS1 fusion assays. In addition, we brought an example of digital PCR (dPCR) into discussion relevant to multiple target detection. We especially discussed recent multiplex-based digital PCR, which successfully applied dPCR to detect multiple targets without sacrificing analytical sensitivity.

We are grateful for your suggestions and comments on our manuscript, which have undoubtedly enhanced its quality. We believe that our responses to your questions will clarify any ambiguous parts of the manuscript. Overall, we are confident that these improvements will significantly contribute to the overall impact of the study.

Round 2

Reviewer 1 Report

Comments and Suggestions for Authors

I appreciate the time and effort you dedicated to addressing my comments and refining the manuscript. Considering complementing  IHC or FISH maybe better.

Comments on the Quality of English Language

Proofreading.

Author Response

Dear Reviewer,

We sincerely appreciate your thorough review of our manuscript titled "A Highly Sensitive XNA-based RT-qPCR Assay for the Identification of ALK, RET, and ROS1 Fusions in Lung Cancer," submitted for consideration to Diagnostics. Furthermore, we are grateful for the opportunity to address the ongoing comments and suggestions provided by the reviewers following the initial round of review.

In response to your suggestions, we have carefully revisited our manuscript and made the necessary revisions. We have provided a detailed point-by-point response to your comments, aiming to address your concerns comprehensively. We hope that these revisions have further strengthened the quality and clarity of our manuscript.

We sincerely thank you once again for your valuable feedback, which has been instrumental in improving our work. We are grateful for the opportunity to refine our manuscript and enhance its contribution to the field of diagnostics.

Sincerely,

Comments and Suggestions for Authors

I appreciate the time and effort you dedicated to addressing my comments and refining the manuscript. Considering complementing IHC or FISH maybe better.

We value your insightful feedback on our manuscript. We appreciate your acknowledgment of the suggestion to include IHC or FISH as complementary experiments, given their established status as gold standard methods for fusion gene detection. As outlined in the introduction section (Lines 74-96), we discuss the advantages and limitations of RT-qPCR and FISH (or IHC) in detecting fusion genes. One notable drawback of FISH or IHC is their ability to examine only one fusion per test. In response, we developed an RT-qPCR assay with the aim of simultaneously detecting multiple fusions, a capability not achievable with FISH. To complement our RT-qPCR results, we validated the results using NGS methodology. Furthermore, we utilized XNA to enhance the sensitivity and specificity of our assays. Given that the NGS method also involves PCR for library generation prior to sequencing, XNA presents an opportunity to increase the sensitivity and specificity of NGS assays. The successful integration of XNA with RT-qPCR underscores its potential for application in NGS methods, particularly targeted NGS assays. Thus, we chose NGS as complementary experiments for validation in this study. While we acknowledge the importance of IHC or FISH, conducting such experiments exceeds our current capabilities. Nonetheless, we remain enthusiastic about exploring these techniques in future research endeavors.

Comments on the Quality of English Language

Proofreading.

We apologize for any oversight in our proofreading process. We have conducted a comprehensive review of the manuscript to address grammatical and spelling errors and have also restructured sentences to enhance clarity. We are confident that these efforts have resulted in significant improvements to the manuscript.

Reviewer 2 Report

Comments and Suggestions for Authors

Many thanks to authors for their detailed response to comments and clear answers to all questions mentioned in the review. All concerns about probe design, percent of cancer cells in samples and information regarding related studies are properly addressed, and no further changes of the manuscript are necessary for its publication.

Author Response

Dear Reviewer,

I would like to extend our heartfelt gratitude for your invaluable support and guidance throughout the review process of our manuscript titled “A Highly Sensitive XNA-based RT-qPCR Assay for the Identification of ALK, RET, and ROS1 Fusions in Lung Cancer," submitted for consideration to Diagnostics.

Your thorough evaluation, insightful comments, and constructive feedback have been immensely beneficial in shaping the final version of our manuscript. Your expertise and dedication to ensuring the quality of scientific research have been truly appreciated.

We are sincerely grateful for your time, effort, and commitment to reviewing our manuscript. Your professionalism and attention to detail have been instrumental in strengthening the quality of our work.

Thank you once again for your invaluable support and acceptance of our manuscript.

Warm regards,

Comments and Suggestions for Authors

Many thanks to authors for their detailed response to comments and clear answers to all questions mentioned in the review. All concerns about probe design, percent of cancer cells in samples and information regarding related studies are properly addressed, and no further changes of the manuscript are necessary for its publication.